# The Process Parameters of Micro Particle Bombarding (MPB) for Surface Integrity Enhancement of Cermet Material and Tool Steel

**DOI:** 10.3390/mi14030643

**Published:** 2023-03-12

**Authors:** Fu-Chuan Hsu, Li-Jie Chen, Zong-Rong Liu, Hsiu-An Tsai, Chin-Hao Lin, Wei-Yu Chen, Hwa-Teng Lee, Tsung-Jen Cheng

**Affiliations:** 1Metal Industries Research and Development Centre (MIRDC), Kaohsiung 81160, Taiwan; 2Department of Mechanical Engineering, National Cheng Kung University, Tainan 70101, Taiwan; 3Power Micro International Co., Limited, Kaohsiung 83051, Taiwan

**Keywords:** micro particle bombarding (MPB), micro blasting, micro shot peening, cermet

## Abstract

In order to increase the performance of tool or mold/die, there are a lot of micro features on the surface to provide special functions, such as anti-adhesion or lubrication. The MPB (Micro Particle Bombarding) process provides a powerful technology to enhance the surface quality without damaging the micro features. The effect of MPB parameters were investigated by bombarding the surface with extremely small particles (20~200 µm in diameter) at a high velocity and pressure to obtain a better surface integrity. -The MPB has two functions, one is micro blasting for cleaning purposes and the other is micro shot peening for surface strengthening. The regression relationship between particle bombarding time and micro hardness is established to predict the surface hardness after MPB process. The experimental results reveal that the surface hardness of cermet is increased 14~66% (HV2167~HV3163) by micro particle bombarding. The micro shot peening provides a good surface integrity due to thebetter surface roughness of 0.1 μmRa and higher compress residual stress of −1393.7 MPa, which is up to 26% enhancement compared with the base material cermet. After micro shot peening, the surface hardness of the SKD11 tool steel increased from HV 686 to HV 739~985. The surface roughness of SKD 11 after micro shot peening was 0.31–0.48 μmRa, while the surface roughness after micro blasting was 0.77–1.15 μmRa. It is useful to predict the residual stress for micro blasting by surface roughness, and to estimate the residual stress after micro shot peening by surface hardness by applying the MPB process in industry in the case of SKD 11 tool steel.

## 1. Introduction

High hardness materials such as hardened steel, tool steel, and tungsten carbidehave been widely used in cutting tools and mold/die to manufacture parts [1]. To increase the rate of production, many factories use high-speed processing, such as high-speed turning and high-speed stamping, but these methods reduce the tool life and mold/die stability. Cutting tools and mold/die in harsh environments were more prone to failure due to wear and fatigue [2], so improving the performance and life of cutting tools and mold/dies has become an indicator of technological development. Fatigue failure of parts mostly occurs on the surface; therefore, surface integrity is an important factor affecting the life of cutting tool and mold/die.

Micro particle bombarding (MPB) was used to improve the life of the mold/die or cutting tools by very small particles bombarding the surface repeatedly at high speed. Figure 1 shows the schematic illustration of MPB on a surface. Compared with conventional shot peening that used a particle size of about 500 µm [3], MPB used an extremely small particle size of about 20~200 µm in diameter. The surface quality improvement of MPB was better than that of conventional shot peening. MPB has two functions, including micro blasting for cleaning purpose and micro shot peening for surface strengthening. Micro blasting/shot peening is a promising technology that can process the various materials with advantages of a high flexibility, no heat affected zone, and a high efficiency [4,5].

With the global trend of net zero carbon emission and carbon tax policy, the production of parts needs to reduce the pollution of waste into the environment. However, the green manufacturing processes, such as dry cutting or high-speed oil-free stamping (above 600 stroke per minute) [6,7] decrease the life of the cutting tools or mold/die significantly. The cutting tools and mold/die need to be replaced frequently, which increases the manufacturing cost. In recent years, sintered materials have been widely used in cutting tools, such as cermet, with ceramic and metal characteristics, it is mostly used in cutting tools due to the high hardness and high temperature resistance [8,9]. SKD 11 tool steel is widely used as a mold/die material. It has great wear resistance and is very suitable for cold work. These materials have good mechanical properties; however, the harsh production environment can easily damage the tools and mold/die [10]. Through MPB technology, the surface defects of the tools and mold/die can be removed to increase their life [11,12].

In previous studies, MPB technology has been applied to many materials, and the effects of different parameters have been investigated to improve the performance of parts, mold/die, and cutting tools [13,14,15,16]. MPB technology can change the microstructure, surface morphology, roughness, residual stress, and fatigue life of the material surface [17,18,19]. Although some research has aimed to study the advantages of micro blasting or micro shot peening, few studies have focused on the prediction of the surface hardness bombarding time. This article aims to investigate the relationship of cermet and SKD 11 between surface integrity (such as surface hardness, roughness, and residual stress) and the process parameters of MPB. The results are useful to enhance the tool and mold/die life applied in industry.

## 2. Materials and Methods

The test materials were high hardness T130A cermet and SKD11 tool steel. The microstructure and chemical composition of SKD 11 steel is shown in Figure 2 and Table 1. The workpiece was machined by a wire electro-discharge machining (Wire EDM). The dimension of the specimen was around 10 mm × 5 mm × 10 mm. The parameter of micro particle bombarding (MPB) included air pressure (lower condition A and higher condition B) and bombarding time, such as 15, 20, 25, and 30 sec, respectively. Figure 3 and Figure 4 show the particle data of micro particle bombarding (MPB).

The microstructure and composition of the material were observed with scanning electron microscope (SEM, HITACHI SU-5000, Japan and Zeiss AURIGA, Germany) and energy-dispersive X-ray spectroscopy (EDS). The surface was photographed and analysed at a working voltage of 20 KV and a magnification of 500×. The phase composition and residual stress were analyzed by a Bruker D8 X-ray diffractometer, the compound phase composition was analyzed by the working voltage of 50 KV, and the residual stress was analyzed by the sin^2^ϴ method. The roughness was measured using a 3D laser confocal microscope (Figure 5, VK-X200 series, Keyence, Japan). The vickers hardness (HV) was used to identify the hardness after micro particle bombarding (MPB).

## 3. Results

### 3.1. Surface Morphology and Microstructure of Cermet

The microstructure of cermet, shown in Figure 6, was revealed by etching with 5 mL distilled Water +4.7 mL HNO_3_ +0.3 ml HCl. According to SEM and EDS analysis, the main elements of chemical composition of cermet are W, Co, Fe, Nb and Ti (Figure 7 and Figure 8). In Figure 7a, the dark gray area is the hard phase of TiCN. In Figure 8a, the light gray area is the binder phase of TiCN. The X-ray diffraction analysis showed that the composition phase of cermet was mainly TiCN mixed with TiC, Co and Ni. (Figure 9). The two phases, TiCN and TiC, provide material strength and hardness that improve cutting tool performance.

### 3.2. Parameters of Micro Particle Bombarding (MPB)

The 3D image of the surface morphology after micro particle bombarding (MPB) treatment is presented in Figure 10. The result of surface roughness is shown in terms of Ra (arithmetical mean roughness value). Through the cross line measurement, the surface roughness was 0.046 µmRa with bombarding pressure A for 15 sec. Figure 11 shows the surface morphology with applied pressure B for 30 sec, and the surface roughness is 0.1 µmRa.

The Vickers hardness indentation after micro particle bombarding (MPB) treatment is presented in Figure 12. Figure 12a shows the indentation shape and the diagonal length were 14 µm and 12.7 µm respectively. The indentation depth was 1.38 µm, as shown Figure 12b.

Figure 13 shows the surface roughness of cermet with different micro particle bombarding times, where the roughness value was increased after micro particle bombarding (MPB). For most conditions, the higher the bombarding time and air pressure, the worse the surface roughness for both the micro blasting (0.36~0.57 μmRa) and micro shot peening (0.046~0.1 μmRa) processes. Micro spherical particles (see Figure 3a,b) were adopted with the function of surface strengthening in the micro shot peening. Micro polygonal particles were used with the purpose of surface cleaning in micro blasting. Therefore, the surface roughness of the micro blasting was higher than that of the micro shot peening.

Figure 14 shows the surface hardness of cermet with different bombarding time, the hardness value was increased after Micro Particle Bombarding (MPB), the surface hardness was improved in the range of 14~66% (see Table 2). The relationship between the surface hardness (Y) and bombarding time (X) are established by regression analysis. The equation is as follows,
Y(Hardness, HV) = 263X(Bombarding time) + 1906(1)

The red dot line expresses the linear Equation (1). This equation is very useful to predict the surface hardness by bombarding time for micro particle bombarding (MPB) with cermet material. Figure 15 shows the cross section of cermet and SKD 11 tool steel after micro particle bombarding (MPB). It is clear that the thickness of surface modification layer for cermet was about 2 μm, and that for the SKD11 tool steel, it was about 5 μm. Figure 16 and Figure 17 show the surface roughness and surface hardness of SKD11 after micro particle bombarding (MPB). Five MPB parameters were selected to evaluate the surface roughness and surface hardness. The surface roughness of SKD 11 after micro shot peening was 0.31–0.48 μmRa, while the surface roughness after micro blasting was 0.77–1.15 μmRa. The initial hardness of SKD 11 was about HV686. After micro shot peening, the surface hardness increased from HV 686 to HV 739~985.

### 3.3. Residual Stress Analysis of Micro Particle Bombarding Process

The residual stresses were analysed mainly for the TiC_0.7_N_0.3_ phase, with the Miller’s index surface (400) as the target, and the sin2θ method was used to analyse the residual stresses of the base material (cermet without MPB process), micro blasting process, and micro shot peening process. The residual stresses of the base material and the micro particle bombarding were all compressive stresses. The residual stress was −1107.5 MPa for the base material. The purpose of micro blasting is surface cleaning, and it induced residual stress at a value of −1297.7 MPa.

The function of micro shot peening is surface strengthening, and therefore the value of residual stress was up to −1393.7 Mpa. The residual stress value of micro blasting was higher than that of the base material, with only 17% enhancement (see Table 3). The parameters of micro blasting and micro shot peening were 30 sec bombarding time with pressure B. The residual stress of micro shot peening was higher than that of micro blasting, which was 26% enhancement compared with the value of the base material. The higher the compress residual stress, the better the fatigue life of the component. From the experiment results of Figure 13 and Figure 18, the micro shot peening process provided a good surface integrity due to the better surface roughness of 0.1 μmRa and higher compress residual stress of −1393.7 MPa.

In the case of the SKD 11 tool steel, the surface compressive residual stress was 5~12% higher than that of the base material after micro blasting. However, the residual stress was 10~22% higher than that of the base material after micro shot peening. The surface compressive residual stress increased by micro shot peening was much higher than that by micro blasting. The test results are shown in Table 4 and Figure 19.

## 4. Discussion

For the same MPB parameters (pressure: B, bombarding time: 30 s), the average surface roughness of cermet was 0.55 μmRa for micro blasting and 0.13 μmRa for micro shot peening. In the case of the SKD 11 tool steel, the average surface roughness was 1.03 μmRa for micro blasting and 0.35 μmRa for micro shot peening, as shown in Table 5. The results revealed that the surface roughness of SKD 11 was about 1.9–2.7 times compared with that of cermet due to the higher hardness of cermet (HV 1906).

The hardness of the base material (cermet) was HV 1906, while it was HV 686 for SKD 11 tool steel. After micro shot peening under the same MPB parameters (Pressure A, bombarding time: 15 s), the surface hardness of cermet increased to HV 2359, and that of the SKD 11 tool steel was increased to HV 789, as shown in Table 6. According to the results, the surface hardness enhancement of cermet was 24%, and that of the SKD 11 tool steel was only 15%. As the hardness of cermet was much higher than for theSKD 11 tool steel, the cermet has a better surface hardness enhancement.

Table 7 shows the enhancement of residual stress for cermet and SKD 11 tool steel under different MPB processes (micro blasting and micro shot peening with the parameter of pressure: B, bombarding time: 30 s). During the MPB process, the grain size of SKD 11 tool steel was relatively coarse, the surface resulted in severe plastic deformation, and induced higher residual stress. In addition, due to the high toughness and ductility of SKD 11 tool steel, a thick hardened layer was formed on the surface (see Figure 15b, about 5 μm), which increased the higher compressive residual stress for SKD 11 tool steel.

In contrast, as the grain size of cermet was relatively fine, the plastic deformation was smaller during the MPB process. Therefore, the surface compressive residual stress of cermet was lower and the hardened layer on the surface of the cermet was thinner (see Figure 15a, only 2 μm). In summary, the enhancement percentage of compressive residual stress for cermet (18%) was higher than that of SKD 11 tool steel (5%) due to the higher hardness of the cermet material.

After micro blasting, the compressive residual stress was about −1109.5~−1389 MPa (see Table 4) on the surface of SKD 11. The linear relationship between the surface roughness and residual stress is shown in Figure 20. The correlation coefficient of the R^2^ value was 0.94, which means it was suitable to predict the residual stress by the value of surface roughness in the case of micro blasting. The relationship between residual stress (Y) and surface roughness (X) was established by regression analysis. The equation for micro blasting is shown as follows,
Y (Residual stress) = −787.92X (Surface roughness, µmRa) − 510(2)

After micro shot peening, the compressive residual stress was about −1203.8~−1490.5 MPa (see Table 4) on the surface of SKD11. The linear relationship was also obtained between the surface hardness and residual stress. In Figure 21, the correlation coefficient was an R^2^ value of 0.94, and it implies that it is reasonable to predict the residual stress using the value of surface hardness in the case of micro shot peening. The relationship between residual stress (Y) and surface hardness (X) was established by regression analysis. The equation for micro shot peening is as follows,
Y (Residual stress) =−1.23X (Surface hardness, HV) − 297(3)

## 5. Conclusions

Higher the bombarding time and air pressure, the worse the surface roughness of cermet for both micro blasting (0.36~0.57 μmRa) and micro shot peening (0.046~0.1 μmRa).The surface hardness of cermet was improved in the range of 14~66% (HV2167~HV3163) by the micro particle bombarding (MPB) process.The relationship between the surface hardness (Y) and bombarding time (X) of cermet was established by regression analysis: Y (Hardness, HV) = 263X(Bombarding time) + 1906. It is very useful to predict the surface hardness using bombarding time for the micro particle bombarding (MPB) process.The micro shot peening process provides a good surface integrity for cermet due to the better surface roughness of 0.1μmRa and higher compressive residual stress of −1393.7 MPa (26% enhancement).The relationship between residual stress (Y) and surface roughness (X) of SKD11 tool steel for micro blasting was established as Y (Residual stress) = −787.92X (Surface roughness, μmRa) −510.The relationship between residual stress (Y) and surface hardness (X) of SKD11 tool steel for micro shot peening was found to be Y (Residual stress) =−1.23X (Surface hardness, HV) −297.

## Figures and Tables

**Figure 1 micromachines-14-00643-f001:**
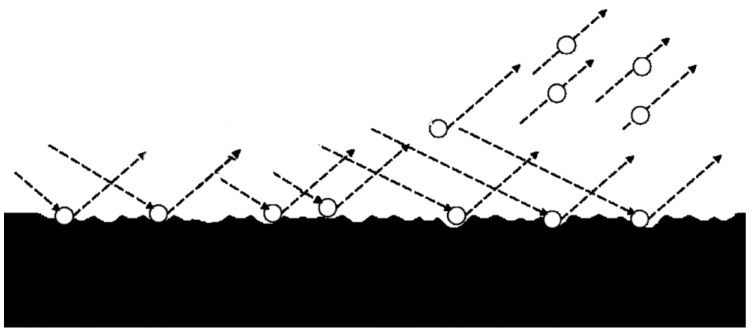
Schematic illustration showing micro particle bombarding (MPB).

**Figure 2 micromachines-14-00643-f002:**
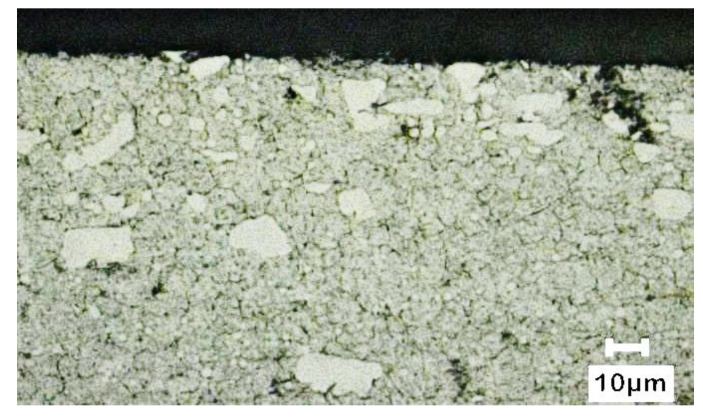
Microstructure of SKD 11 tool steel.

**Figure 3 micromachines-14-00643-f003:**
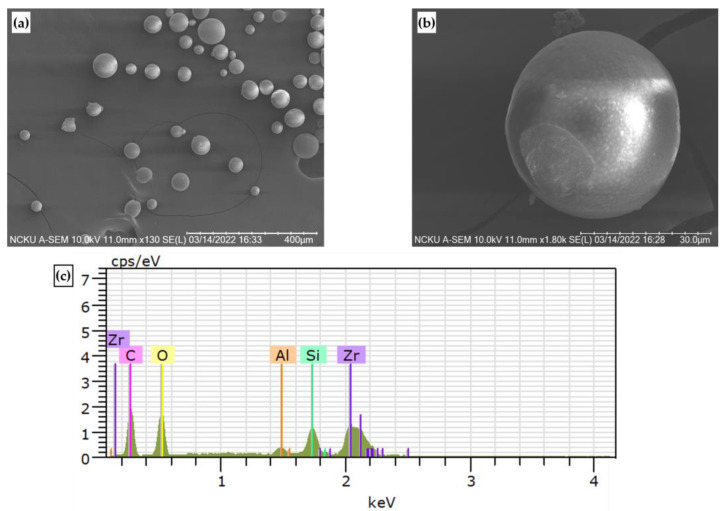
SEM image and EDS image of micro shot peening shots: (**a**) 130×, (**b**) 1800×, and (**c**) EDS.

**Figure 4 micromachines-14-00643-f004:**
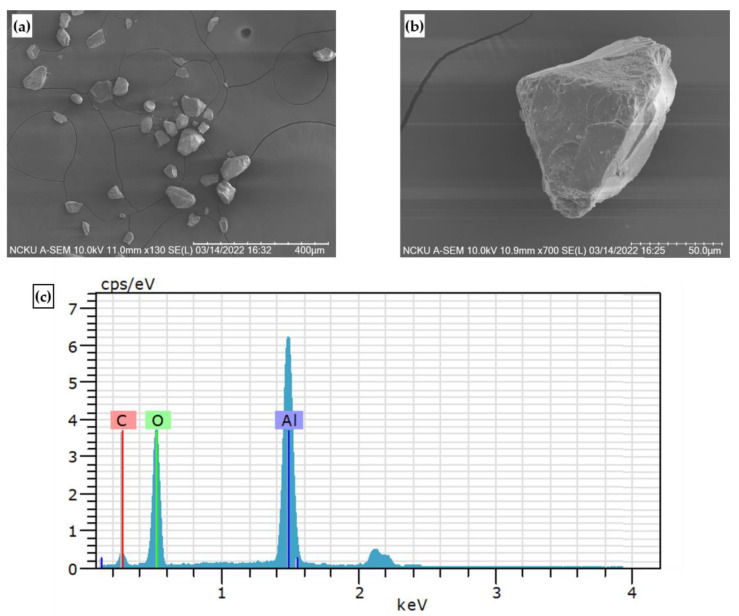
SEM image and EDS image of micro blasting abrasive: (**a**) 130×, (**b**) 700×, and (**c**) EDS.

**Figure 5 micromachines-14-00643-f005:**
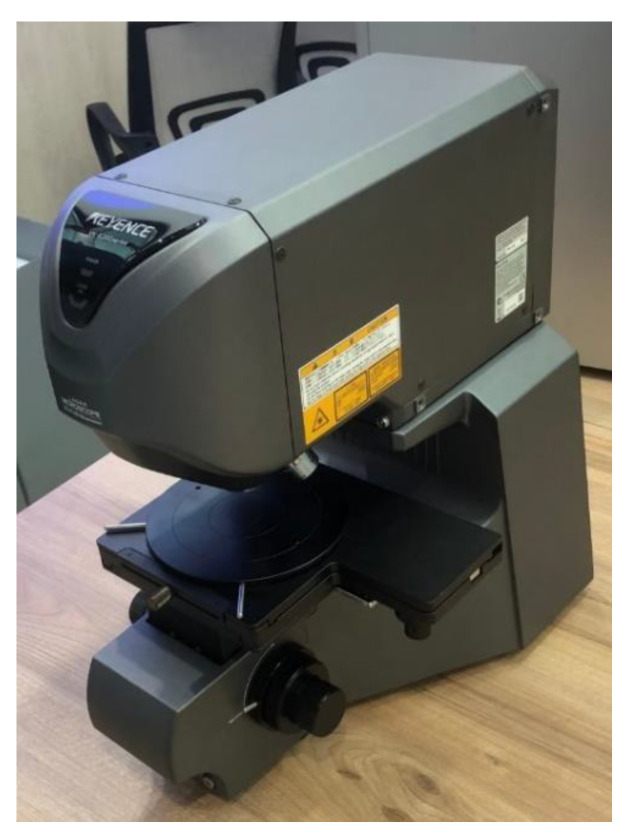
3D laser confocal microscope (VK-X200 series, Keyence, Japan).

**Figure 6 micromachines-14-00643-f006:**
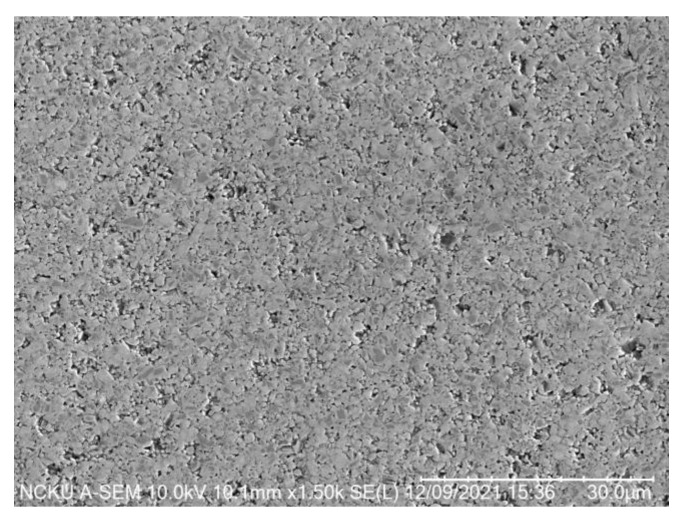
Surface morphology of cermet (SEM).

**Figure 7 micromachines-14-00643-f007:**
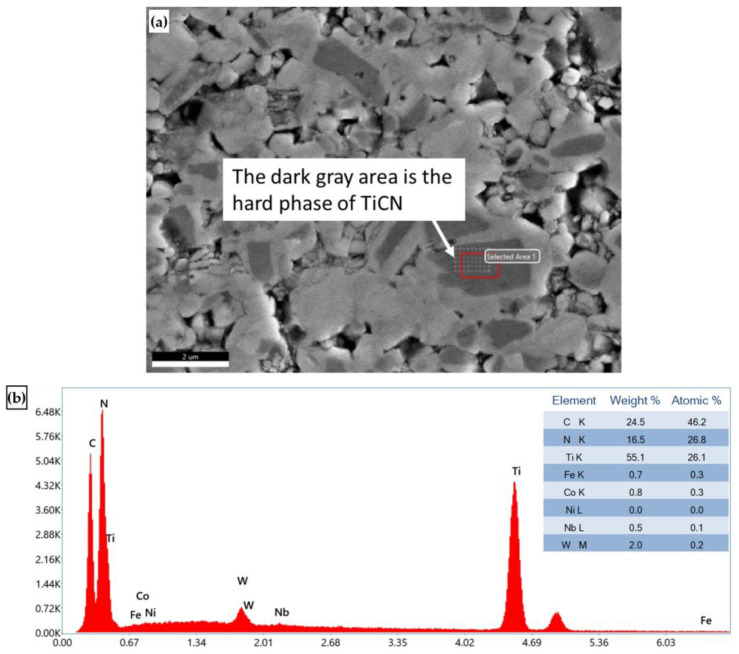
(**a**) SEM micrographs and (**b**) EDS microanalysis of cermet hard phase.

**Figure 8 micromachines-14-00643-f008:**
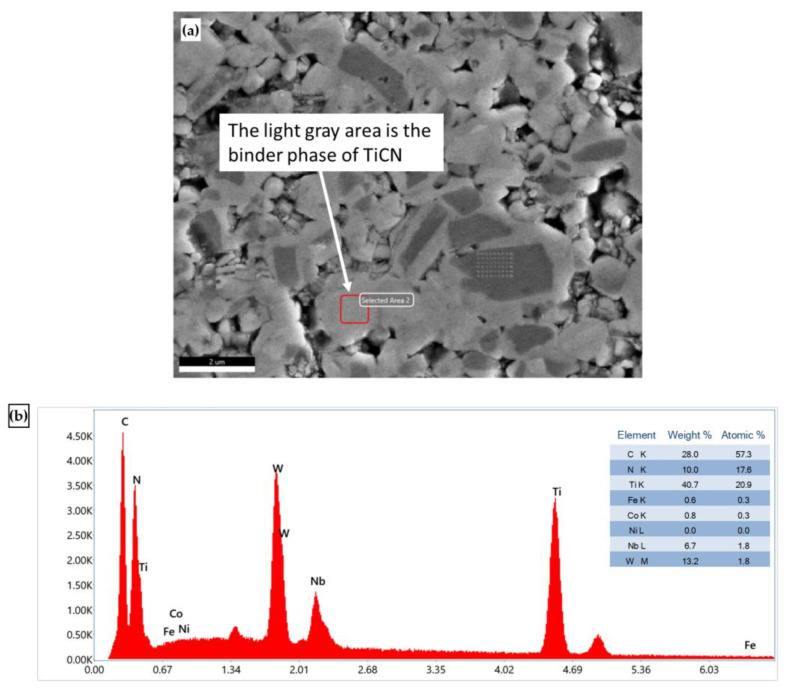
(**a**) SEM micrographs and (**b**) EDS microanalysis of cermet binder phase.

**Figure 9 micromachines-14-00643-f009:**
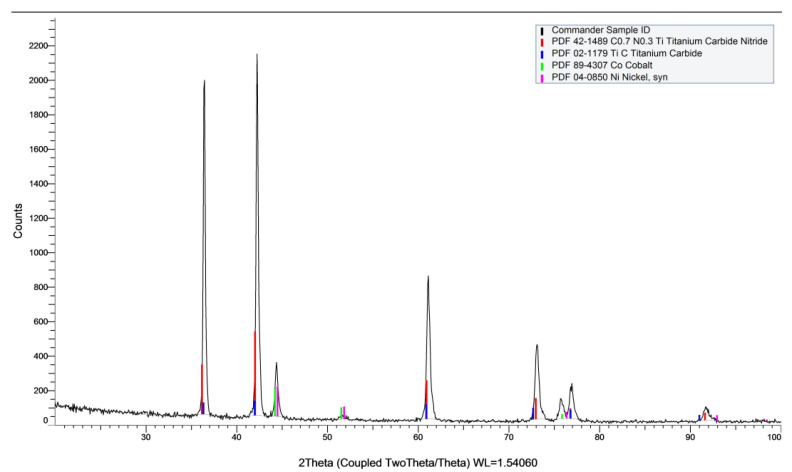
X-ray diffraction analysis of the chemical compound composition of T130A cermet.

**Figure 10 micromachines-14-00643-f010:**
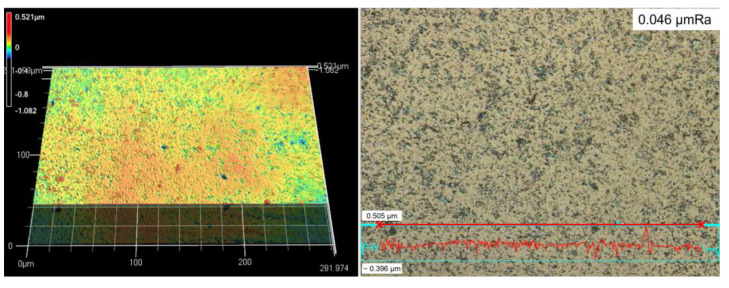
The surface roughness of cermet after MPB at lower pressure A for 15 s (0.046 μmRa, micro shot peening).

**Figure 11 micromachines-14-00643-f011:**
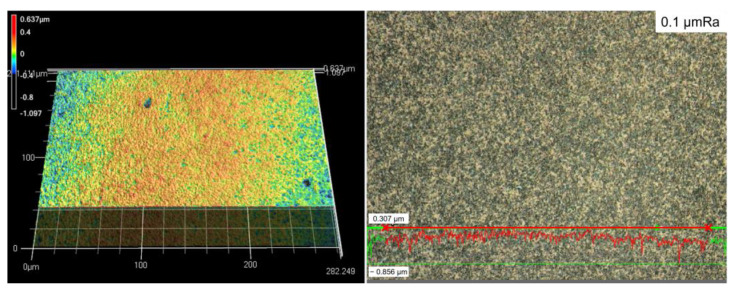
The surface roughness of cermet after MPB at higher pressure B for 30 s. (0.1 μmRa, micro shot peening).

**Figure 12 micromachines-14-00643-f012:**
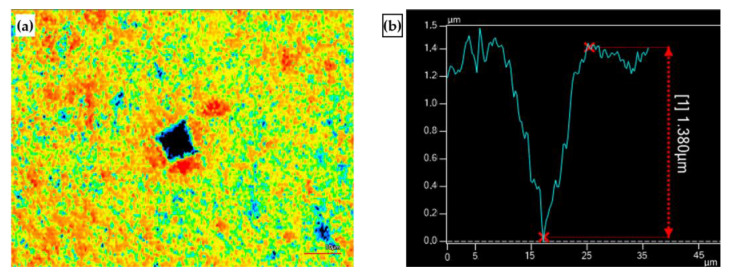
The Vickers hardness indentation of cermet on lower pressure A of micro shot peening for 15 s (load: 200 g). (**a**) The Vickers hardness indentation morphology. (**b**) The Vickers hardness indentation depth.

**Figure 13 micromachines-14-00643-f013:**
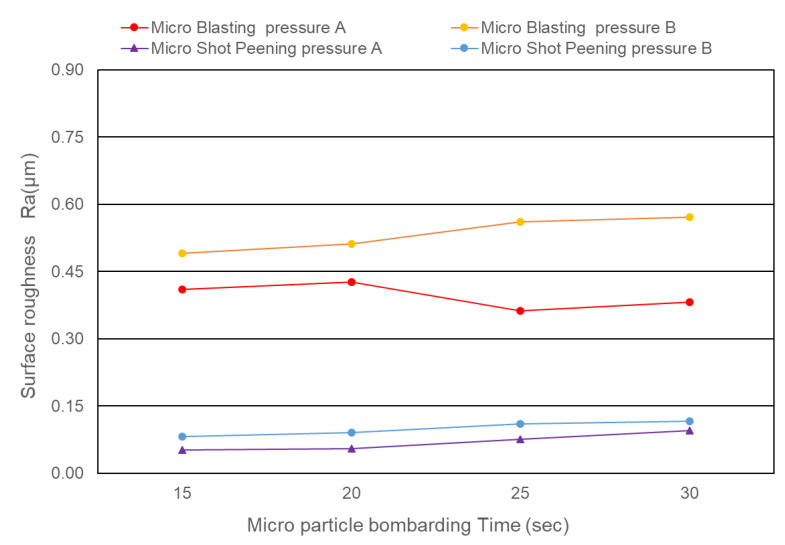
The surface roughness of cermet after micro blasting (about 0.45 μmRa) and micro shot peening (about 0.07 μmRa) for different bombarding time and pressure.

**Figure 14 micromachines-14-00643-f014:**
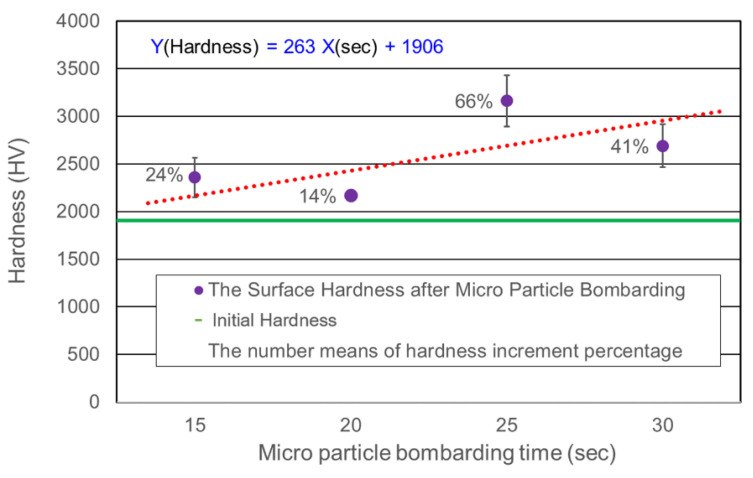
The surface hardness of cermet after micro particle bombarding (MPB) with different bombarding times for micro shot peening process.

**Figure 15 micromachines-14-00643-f015:**
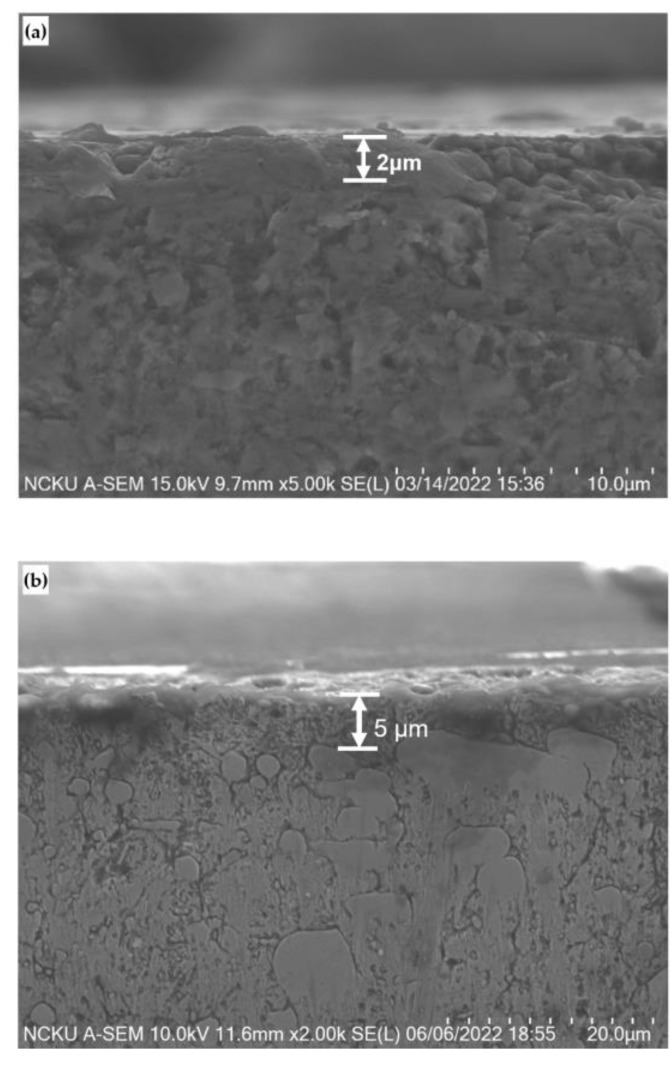
The cross section of cermet and SKD 11 tool steel after micro particle bombarding for (**a**) cermet and (**b**) SKD 11 tool steel.

**Figure 16 micromachines-14-00643-f016:**
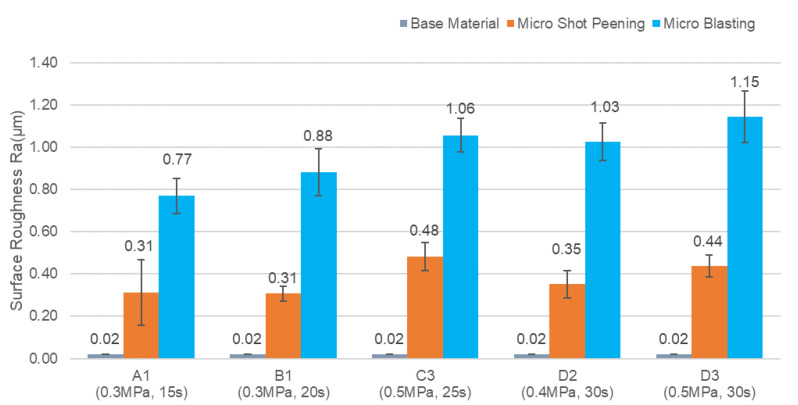
The surface roughness of SKD 11 after micro particle bombarding (MPB).

**Figure 17 micromachines-14-00643-f017:**
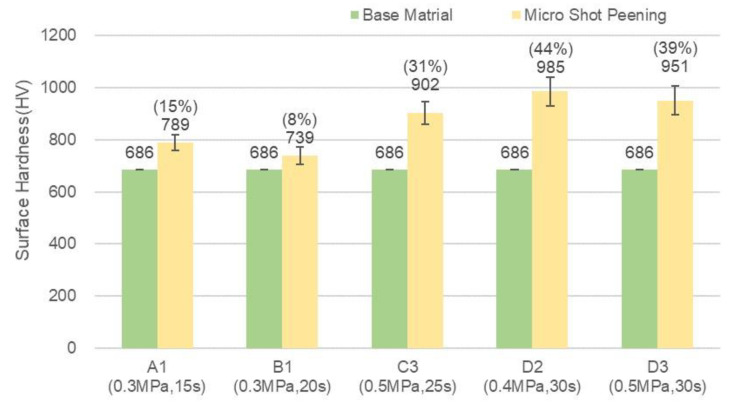
The surface hardness of SKD 11 after micro particle bombarding (MPB).

**Figure 18 micromachines-14-00643-f018:**
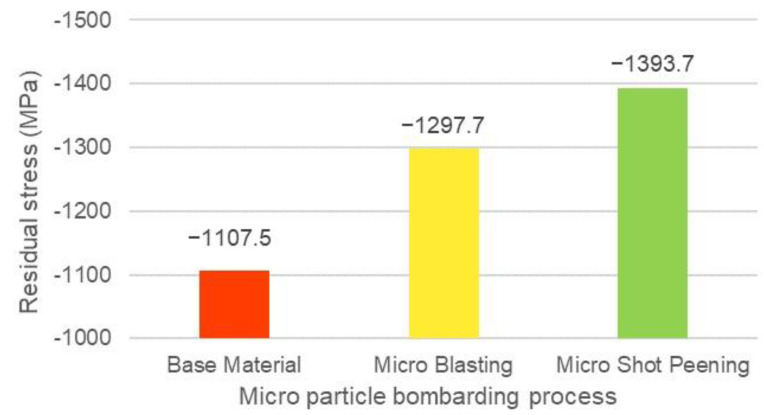
Residual stress analysis of cermet for the micro particle bombarding process.

**Figure 19 micromachines-14-00643-f019:**
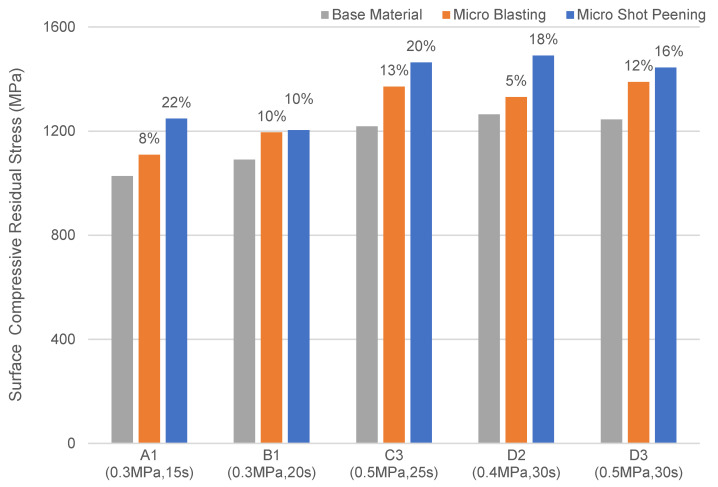
Surface compressive residual stress of SKD 11 under different micro particle bombarding (MPB) parameters.

**Figure 20 micromachines-14-00643-f020:**
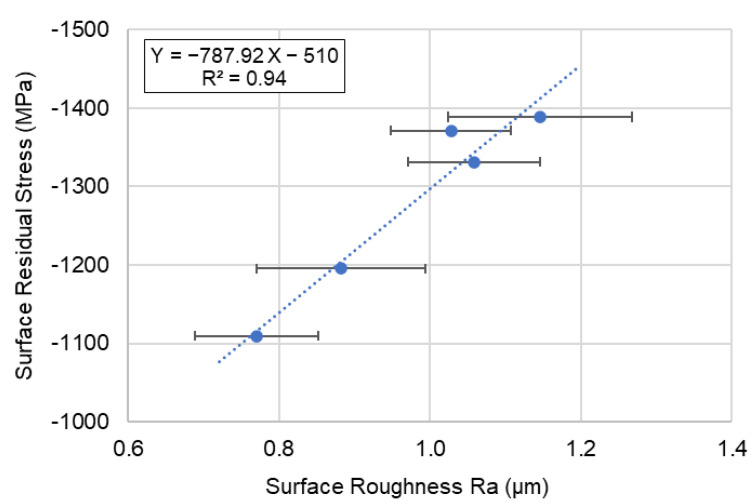
Relationship between surface roughness and surface residual stress for micro blasting (SKD11 tool steel).

**Figure 21 micromachines-14-00643-f021:**
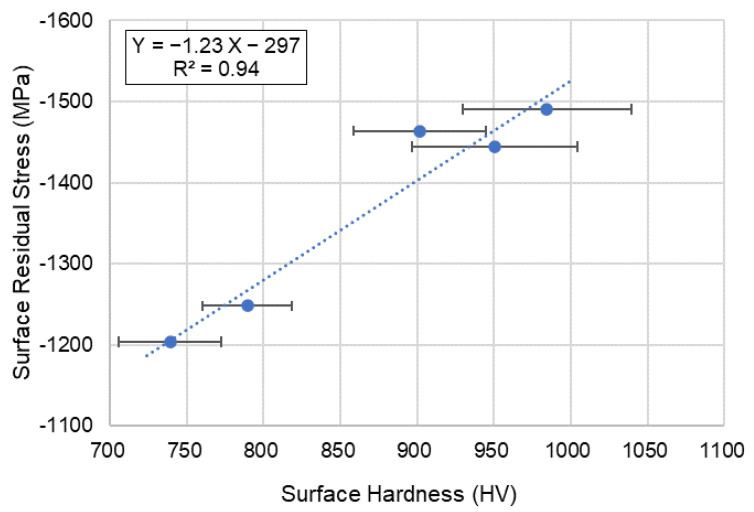
Relationship between surface hardness and surface residual stress for micro shot peening (SKD11 tool steel).

**Table 1 micromachines-14-00643-t001:** The chemical composition of SKD 11 tool steel.

Material	Chemical Composition(wt%)							
	C	Si	P	S	Mn	Cr	Mo	Ni
SKD 11	1.4–1.6	0.1–0.6	0.03	0.03	0.1–0.6	11.0–13.0	0.82	-

**Table 2 micromachines-14-00643-t002:** The hardness enhancement of cermet after micro shot peening.

Bombarding Time (s)	0	15	20	25	30
Hardness (HV)	1906	2359	2167	3163	2689
Increment (%)	--	24%	14%	66%	41%

**Table 3 micromachines-14-00643-t003:** The residual stress of cermet enhancement after micro particle bombarding (MPB).

Micro Particle Bombarding (MPB)	Base Material	Micro Blasting	Micro Shot Peeing
Residual Stress (MPa)	−1107.5	−1297.7	−1393.7
Increment (%)	-	17%	26%

**Table 4 micromachines-14-00643-t004:** The residual stress of SKD11 by micro particle bombarding (MPB).

	Processes	Base Material (MPa)	Micro Blasting (MPa)	Micro Shot Peening (MPa)
MPB Parameters				
A1 (0.3 MPa, 15 s)		−1027.1	−1109.5	−1248
B1 (0.3 MPa, 20 s)		−1090.6	−1195.3	−1203.8
C3 (0.5 MPa, 25 s)		−1218.6	−1371.2	−1463.8
D2 (0.4 MPa, 30 s)		−1264.6	−1331.3	−1490.5
D3 (0.5 MPa, 30 s)		−1244.7	−1389	−1444.5

**Table 5 micromachines-14-00643-t005:** The surface roughness after MPB for cermet and SKD 11 tool steel.

	Material	Cermet	SKD 11 Tool Steel	Increase Ratio (SKD 11/Cermet)
Process				
Micro blasting		0.55 μmRa	1.03 μmRa	−1.9
Micro shot peening		0.13 μmRa	0.35 μmRa	2.7

**Table 6 micromachines-14-00643-t006:** The surface hardness enhancement for cermet and SKD 11 tool steel.

	Process	Base Material	Micro Shot Peening	Hardness Enhancement (%)
Material				
Cermet		HV 1906	HV 2359	24%
SKD 11 tool steel		HV 686	HV 789	15%

**Table 7 micromachines-14-00643-t007:** The residual stress enhancement for cermet and SKD 11 tool steel.

	Material	Cermet (MPa)	Cermet Enhancement (%)	SKD 11 (MPa)	SKD 11 Enhancement (%)
Process					
Base material		−1107.5	-	−1264.6	-
Micro blasting		−1297.7	17%	−1331.3	5%
Micro shot peening		−1393.7	26%	−1490.5	18%

## Data Availability

The data presented in this study are available on request from the corresponding author.

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
