# Peer review of "The Process Parameters of Micro Particle Bombarding (MPB) for Surface Integrity Enhancement of Cermet Material and Tool Steel"

_micromachines, 2023, doi:10.3390/mi14030643_

Round 1

Reviewer 1 Report

Paper "The Process Parameters of Micro Particle Bombarding (MPB) for Surface Integrity Enhancement of Cermet Material and Tool Steel" by Fu-Chuan Hsu, Li-Jie Chen , Zong-Rong Liu , Hsiu-An Tsai, Chin-Hao Lin, Wei -Yu Chen, Hwa-Teng Lee and Tsung-Jen Cheng touches on the current topic of surface treatment. Micro-bombarding is one of the methods for reducing surface roughness, however, the processes that occur with treated surfaces are rarely touched upon in research. The authors of this article conducted research on two materials - cermet and tool steel. These materials are often used in the manufacture of cutting tools. As you know, a decrease in surface roughness leads to a decrease in the wear rate of the tool material, a decrease in energy losses, etc. In addition to measuring the surface roughness of processed materials, the authors also studied such parameters as hardness and residual stresses.

However, despite the scientific significance of the research, there are a number of remarks:

1. Line 15 extra dot after "is ."

2. Figures 3, 4 add scale bar and labels of structural constituent materials.

3. Line 127 figures 9, 10, 12. The term "micro shot peening" is used, previously "micro peening" was used. It is necessary to bring the text to common terms.

4. Figure 13 "Initial hardness" - correct "Initial hardness"

5. Usually, the term "Micro-bombarding" or "microbombarding" is used, as well as "micro-penning" (instead of "Micro bombarding", "micro penning").

6. There is not enough figures or scheme of surface treatment. There are also no images of the test-bench itself.

7. There is no data on the particles that are processed (material, size, hardness, shape, etc.)

8. There is a lack of microstructure and chemical composition of SKD 11 steel.

9. To what surface depth does SKD 11 increase in hardness?

10. It is not clear from the text when the process of micro-penning occurs, and when micro-blasting occurs. What is the difference between these processes. The text of the article does not contain information on how the processing time of ceramic materials affects to residual stresses.

11. Change in hardness depending on the processing time, it is better to transfer to section 3 - Results.

12. Why is there only one dependence for ceramic materials, and two for SKD 11 steel? Perhaps it is worth making the same dependences for ceramic materials and for steel, for example, "hardness - processing time"; "residual stresses - processing time"; "roughness - processing time", etc.

13. The conclusions and discussion should describe the identified differences between the processing of steel and ceramics.

14. The list of literature should be expanded and more information about research in this area should be added.

Author Response

Dear reviewer, 

Thank you very much for your suggestions to make this paper become better.

Best wishes,

Fu-Chuan Hsu

MIRDC/Taiwan

Reviewer 2 Report

Dear Authors, thanks for your submittion.
The artical' scientifif soudness and impact is eonugh. But the article can be strengthened when attention is paid to the following elements:

- Insufficient number of citations. More references strengthen the publication. Citations can be selected from the last 5 years and IF higher journals.

- Authors can limit self-citation.

- The literature part seems weak.

- Further clarification in the results is needed.

- The evaluation and conclusion section needs further clarification.

In this paper has been discussed in which parameters Micro Particle Bombardment will be successful for Improving the Surface Integrity of tool steels of Cermet Material.

The work is partially original. The parameters studied and the conclusion of the method can contribute to the literature. It is a work in the field.

Surface roughness is a subject studied. Microparticle bombardment is a well-known method. The experiments and results applied here can be used and evaluated in the field. 

The scientific definition can be made more specific in order to attract the attention of the reader. The title section can help with this. The scientific background of the study and which problem it directly focuses on should be stated in the abstract. The description of the method, its reference and results can be explained with specific examples in the last section.

The problems, methods and results discussed show internal consistency.

It may be appropriate to select references from journals that are directly related to the subject and have high impavt factor within the last year.

Author Response

(The authors gave the same response as above.)

Reviewer 3 Report

In this paper, the surface roughness, micro hardness and residual stress after micro particle bombarding were investigated. The idea seems interesting. However, there are still many problems remaining to be solved.

1. The authors are suggested to give more content on the innovation of this paper in abstract.

2. Why were two different materials (cermet, SKD11) used in this paper?

3. To improve the surface roughness, micro hardness and residual stress of cermet material, there are many methods can be used. Why was micro particle bombarding selected in this paper?

4. What other people have done for the methods and materials involved in this article? It is suggested that the authors conduct more in-depth literature review.

5. It is suggested to supplement pictures of experimental materials, MPB equipment and measurement equipment.

6. Many pictures in this paper are not clear enough. The authors are suggested to replace them with clearer ones.

7. In fig. 8, the curve colors for micro blasting pressure A and micro peening pressure B are too similar. The authors are suggested to replace one of them.

8. The authors should emphasize the innovation of the paper in conclusions.

Author Response

(The authors gave the same response as above.)

Round 2

Reviewer 1 Report

All comments have been corrected. The paper is recommended for publication

Reviewer 2 Report

Aceept it as

Reviewer 3 Report

This version is significantly improved, and all main issues have been addressed. This work could be considered for publication as it is.